# Magnitude, resistance profiles, and risk factors of intestinal parasites and enteric bacteria among food handlers in West Guji Zone, Ethiopia

**Lechisa Asefa**[1]*, **Gudeta Kumela**[1], **Habtamu Roba**[1], **Angefa Ayele**[2], **Gedeno Karbana**[1], **Degefa Dhengesu**[1], **Hailu Lemma**[1], **Anteneh Fikrie**[2]

**1** Lecturer at the Department of Environmental Health, Institute of health, Bule Hora University, Bule Hora Town, Ethiopia, **2** Lecturer at the Department of Public Health, Institute of Health, Bule Hora University, Bule Hora Town, Ethiopia

* lechisa123@gmail.com

## Abstract

### Introduction

Despite food handlers' crucial role in pathogen transmission, current data on intestinal parasites and antimicrobial susceptibility and resistance patterns of enteric bacteria among food establishment workers in Bule Hora town are notably limited. This knowledge gap hinders effectiveness of public health interventions.

### Objective

This study aimed to assess the prevalence and antimicrobial resistance profile of intestinal parasites, enteric bacteria, and associated factors among food handlers in Bule Hora town food and drinking establishments.

### Methods

We employed a cross-sectional study design to assess the health status of food handlers. Data were collected utilizing both a semi-structured questionnaire to gather self-reported information and an observational checklist to assess practices and environmental factors. A total of 375 stool samples were collected from the participating food handlers. These samples underwent laboratory examination for the identification of bacterial and parasitic pathogens. To determine factors independently associated with the health status of food handlers, multivariable logistic regression analysis was performed. Adjusted odds ratio together with 95% Confidence Interval (CI) were used to measure the association and p value < 0.05 was used to declare the statistical significance.

### Result

Overall, 147(39.2% (95% CI: 34.26, 44.14)) prevalence of intestinal parasites and 50 (13.3%) (95%CI: 9.2–19.7) prevalence of bacteria was found in this study. The

which permits unrestricted use, distribution, and reproduction in any medium, provided the original author and source are credited.

**Data availability statement:** All relevant data are within the paper and its Supporting Information files.

**Funding:** The author(s) received no specific funding for this work.

**Competing interests:** The authors have declared that no competing interests exist.

**Abbreviation:** KIA, klinger iron agar; LIA, lysine iron agar; EFMHCA, Ethiopia food medicine and health care authority.

most predominant parasite was *Ascaris lumbricoid (51.1%)* and the most frequently reported an enteric bacterium was *Salmonella* 39 (78%). Out of 39 *Salmonella* isolates, 20.5% showed multidrug resistance, while none of the 11 *Shigella* isolates were multidrug resistant. From multivariable binary logistic analysis, unable to read and write (AOR = 1.96, 95% CI 1.18, 3.26), not trimming fingernails (AOR = 2.52; 95% CI: 1.20, 9.55), not washing hand with soap after toilet (AOR = 2.27, 95% CI 1.17, 4.42), and eating raw food (AOR = 1.62, 95% CI 1.39, 5.00) have a significant association with the prevalence of bacteria and parasites.

## Conclusion

Intestinal parasites and bacteria were among the leading causes of morbidity in the study area. Educational status, eating raw vegetables or fruit, hand washing after the toilet, and fingernail trimming show significant associations with the health status of food handlers.

## Background

In this century, foodborne diseases have become a public health problem around the globe [1]. Worldwide, 10% of people become ill due to eating contaminated food, with the greatest impact observed in Africa and Southeast Asia [2]. Food-borne diarrheal disease kills an estimated 2 million people annually [3]. In Ethiopia, even though there is limited data, there were 280,458 out-patient cases of foodborne disease, which account for 9.39% of all cases [4]. In addition, food-borne illness results in estimated economic costs in terms of pain and suffering, reduced productivity, and medical expenses is substantial, in the range of $10–83 billion each year [5].

Various studies indicate that the primary causes of intestinal parasitic infections in Ethiopia include *Ascaris lumbricoides, Entamoeba histolytica, Giardia lamblia, Trichuris trichiura, and hookworm* [6]. Worldwide gastroenteritis which is caused by *Salmonella and Shigella* species is the most common health problem. Similarly in Ethiopia *Salmonella, Shigella, Listeria monocytogenes, and Escherichia coli (E. coli)*, are among the most frequently reported food borne pathogens, with *Salmonella and Shigella* showing the highest prevalence [7–9]. In Ethiopia, Salmonella accounts for up to 57.9% of foodborne diseases, while Shigella accounts for 10–15% [7].

Besides, *Salmonella* and *Shigella* are major enteric pathogens whose increasing antimicrobial resistance poses a significant global public health challenge [8]. Both organisms commonly exhibit resistance to first-line antibiotics such as ampicillin, chloramphenicol, and trimethoprim–sulfamethoxazole, making empirical treatment increasingly difficult [9]. Consequently, antimicrobial resistance in *Salmonella* and *Shigella* results in prolonged illness, increased hospitalization, and reduced efficacy of commonly used therapies.

Food handlers infected with intestinal parasites and bacteria are potential sources of infections for customers directly or indirectly through food, water, nails, and fingers from food handlers [10]. Whereas, In developing countries, due to poor in

implementation of regulation for food hygiene, food handlers are appointed in food and drinking establishments without a medical certificate on their health status for the common intestinal parasite [10]. The majority of asymptomatic individuals with parasitic infections can be considered as potential source of infection to society because such food handlers routinely practice their jobs without paying attention to the transmission of infections.

Although food handlers are the primary source of infection in food and drinking establishments and this day antimicrobial resistance shaking the globe and becoming alarming issues the prevalence of intestinal parasite, Salmonella, Shigella species, and their antimicrobial susceptibility patterns in Ethiopia is limited. Thus, this study aimed to assess the prevalence and antimicrobial profile of *Salmonella, Shigella,* and intestinal parasites, and their determinant factors among food handlers in food and drinking establishments in Bule Hora town.

## Methods and materials

### Study design, area, and period

A facility-based cross-sectional design study was conducted in Bule Hora town from November to December 2023. Bule Hora is a town in southern Ethiopia, located on the paved Addis Ababa-Moyale highway in the west Guji zone of Oromia. It is the largest town in this zone, mainly inhabited by the Guji Oromo. It has a latitude and longitude of 5° 35′ N and 38° 15′ E and an altitude of 1716 meters above sea level. It is located 467 kilometers away from Addis Ababa, toward the south. The number of food and drinking establishments in Bule Hora town was 273, with 1367 food handlers.

### Population, sample size determination, and sampling procedure

The study population of this study was selected food handlers among food and drinking establishments in Bule Hora town. The sample size was calculated using a single population proportion formula with the assumptions of a 95% confidence interval and a 5% marginal error.

$$n = (Z1 - \alpha/2)^2\, P(1 - P)/d2,$$

Where "n" is the required sample size. P- Proportion, d-margin of error.

Taking p = 44.6% (0.446) (the prevalence of intestinal parasites among food handlers in Metu town, south-west Ethiopia) [11] was used after assessing the proportion that could give the maximum sample size from all variables under study. The final sample size was 416, including a 10% non-response rate.

This sample size was proportionally allocated to all food and drink serving establishments. Then, participants were selected by a simple random sampling technique using the lottery method. The selected food handler was observed for an assessment of personal hygiene, interviewed to assess practice, and examined for the presence of food-borne pathogens.

### Data collection tool, procedure, quality assurance, and laboratory processing

A standardized and structured questionnaire and observational checklist, which were adopted from EFMHCA guidelines and reviewed literature [8,10], were used for the purpose of data collection. Data was collected through face-to-face interviews and observation. The stool was collected in clean, dry, leak-proof stool cups from each selected food handler. The stool samples were transported for examination immediately after collection. For enteric bacteria, approximately 1 gram of stool samples was transferred immediately into the Cary Blair transport medium, labeled, and transported within one hour of collection in the ice-packed cold box (4 °C) to the examination area and analysis was done within 48 hours of collection [12]. Medical laboratory technologists and environmental health professionals with a minimum of a BSc degree were used as data collectors.

To ensure data quality, training was given to data collectors. Daily supervision was provided by the supervisor during the data collection. The questionnaires was pretested with a pilot survey of a similar study population (Yabello) prior to

the actual data collection period. The questionaries' language was changed to the local language. Data consistency was assured throughout data collection, entry, and analysis.

The specimen was checked for serial number, quality, and procedures of collection. The obtained sample was taken to the laboratory immediately after collection. Experienced laboratory professionals were involved in the culture and microscopic examination of the stool. The sterility of each medium was checked by incubating it overnight at 37 °C. The media that showed growth were discarded and replaced by new media that didn't show any growth following overnight incubation. The reference of American-type culture collection strains such as E. coli (ATCC 25922) and P. aeruginosa (ATCC 27853) was used as quality control throughout the study for culture and antimicrobial susceptibility tests.

### Microscopic examination of stool

The stool was observed macroscopically for the presence of adult stages of some intestinal helminthes, consistency, and color. After visual assessment of the stool, intestinal parasites were examined microscopically from each stool specimen using both direct smears mounted in saline and the formal ether concentration procedure as recommended [13].

### Culture and identification of *salmonella and shigella*

A loop-full fecal suspension was added to the selenite F broth and incubated at 37 °C for 18 hours. After incubation, a loop full of the suspension colonies was inoculated onto Xylose Lysine Deoxycholate Agar (High Media, India). Then it was incubated aerobically for 24 hours at 37 °C. After overnight incubation, the growth of *Salmonella* and *Shigella* species was detected by their characteristic appearance on XLD agar (*Salmonella*: red with or without the black center, *Shigella*: red or pink colonies). Five series of biochemical tests, including Klinger iron agar (KIA), Simmons citrate agar, sulfide indole motility test (motility, H2S production, indole), lysine iron agar (LIA), and urease test, were used for the final identification of the bacterial isolate [14].

### Antimicrobial susceptibility test

Antimicrobial susceptibility testing was performed using the Kirby-Bauer disk diffusion method as described by the Clinical Laboratory Standards Institute [15–17]. The pure culture was transferred to a tube containing 5 mL of sterile normal saline (0.85% NaCl) and mixed gently until it formed a homogeneous suspension. The turbidity of the suspension was adjusted to an optical density equivalent to 0.5 McFarland standards. A sterile cotton swab was then dipped into the suspension, and the excess was removed by gentle rotation of the swab against the surface of the tube. The swab was distributed evenly over the entire surface of the Mueller-Hinton agar (Oxoid, UK). The inoculated plates were left at room temperature to dry for 3–5 minutes. Six antimicrobial discs (Oxoid, UK): ampicillin (10 μg), trimethoprim-sulfamethoxazole (23.75/1.25 g), ciprofloxacin (5 μg), gentamicin (10 μg), ceftriaxone (30 μg), and doxycycline (30 μg) were placed aseptically on the inoculated plate using sterile forceps. After 24-hour incubation at 37 °C, the zone of inhibition, including the disk, was measured using a digital caliper to the nearest whole millimeter and interpreted as sensitive, intermediate, or resistant, based on interpretive breakpoints [13,14].

### Study variables and operational definitions

**Study variable.** Our Dependent variable was the prevalence of bacteria or parasites, and the independent variables in this study were Socio-demographic data (sex, age, educational status, marital status, income, training status, etc.) and Hygienic practice (hand washing after toilet, finger trimming, wearing jewelery, eating raw food, etc.)

**Operational definition.**

**Food hygiene practice level:  Practices:** To assess the level of practices, respondents were asked 11 questions from the questionnaire, and those who scored ≤ the mean value were considered to have poor practices, and those who scored > the mean value were considered to have good practices.

**Data processing and analysis.** The collected data were entered into Epidata version 3.1 and exported to SPSS version 26.0 for data cleaning and analysis. Bivariate logistic regression was used to analyze the data, and variables with a p-value of ≤0.25 were selected for the multivariable logistic regression analysis. Multivariable logistic regression was computed to identify factors associated with the health status of food handlers. The variables with a p-value < 0.050 were taken as statistically significant and associated with the sanitation, and hygiene of FDEs. Model fitness was checked by Hosmer and Lemeshow goodness, and it was 0.76 and multi-collinearity was checked with the variance inflation factor.

## Ethical clearance

Ethical clearance to undertake the study was obtained from the Bule Hora University Institutional Research Ethics Review Committee (IRERC) (Reference Number: BHU/IOH/IRERC/017/15). The data were collected after obtaining informed written consent from the study participant. All the study participants were informed about the purpose of the study and their right to refuse and withdraw from study at any time. All the methods have been performed following Declaration of Helsinki. Moreover, information regarding any specific personal identifiers, such as the names of the participants, was not collected and confidentiality of any personal information was also maintained.

## Result

### Socio-demographic characteristics of food handlers

This study enrolled 375 food handlers, achieving a 90.14% response rate. The participants' ages ranged from 18 to 56 years. A majority, 221 (59.2%), had 1–5 years of experience as food handlers. Regarding education, 270 (72.4%) was literate. Notably, 229 (61.4%) of the food handlers had not received formal training in proper food handling practices within their establishments (Table 1).

### Assessment of food handler hygiene practices

The study revealed significant deficiencies in food handler practices. A substantial majority, 268 (71.5%), often neglected hand-washing before handling food, and 235 (62.6%) inconsistently washed hands after toilet use. A critical finding was the lack of detergents at hand washing facilities for 297(79.3%) of handlers. While about half (51.6%) covered their hair and 53.7% wore gowns, a concerning 59.5% of gowns were unclean. Furthermore, 194 (51.6%) wore jewellery and 192 (51.2%) had untrimmed fingernails. Although 293 (78%) reported taking sick leave when ill, risky practices like money handling 247 (65.9%) and eating raw foods 157 (41.9%) were common. Overall, the assessment indicated poor hygiene practices in 303 (81.9%) of the food handlers (Table 2)

### Prevalence of intestinal parasite

Of 375 stool samples examined, 147(39.2% (95% $_{CI}$: 34.26, 44.14)) stool specimens were positive for different intestinal parasites. Among intestinal parasites, *Ascaris lumbricoides* was the most frequently observed parasite, found in 71 individuals, accounting for 51.1% of the total sample. Following this, *Trichuris trichiura* was detected in 29 individuals, making up 20.9% of the cases. The least common parasite observed was *Giardia lamblia*, affecting 5 individuals, or 3.6% of the sample (Table 3).

### Prevalence of salmonellosis and shigellosis

Of 375 stool samples cultured for *Salmonella* and *Shigella*, 50 (13.3%) (95%CI: 9.2–19.7) food handlers were positive. *Salmonella* was isolated from 39 (78%) positive samples for enteric bacteria, and the rest were positive for *Shigella.*

**Table 1. Socio-demographic characteristics of food handlers in West Guji Zone, Ethiopia, 2023. (n = 375).**

| Characteristics | | Frequency | Percentage (%) |
|---|---|---|---|
| Sex | Male | 90 | 24.4 |
| | Female | 285 | 76.1 |
| Age in years | <20 | 95 | 25.5 |
| | 21-30 | 201 | 53.9 |
| | 31-40 | 56 | 15.0 |
| | 41-50 | 21 | 5.6 |
| | ≥ 51 | 2 | 0.5 |
| Marital status | Single | 145 | 38.9 |
| | Married | 206 | 55.2 |
| | Divorced | 14 | 3.8 |
| | Widowed | 10 | 2.7 |
| Educational status | Cannot read & write | 182 | 49.6 |
| | Read & write | 193 | 51.4 |
| Income (in ETB) | ≤ 2000 | 144 | 38.6 |
| | ≥2000 | 231 | 61.9 |
| Service years | ≤1 | 104 | 27.9 |
| | 1-5 | 221 | 59.2 |
| | ≥ 5 | 50 | 13.4 |
| License status of food service establishment | Licensed | 263 | 70.5 |
| | Not licensed | 112 | 30.0 |
| Received training in proper food handling practices | Yes | 187 | 49.8 |
| | No | 188 | 50.2 |

## Antimicrobial susceptibility patterns of *Salmonella* and *Shigella* isolates

Thirty-nine *Salmonella* and eleven *Shigella* were isolated and tested against eight antimicrobial agents; eight isolates of *Salmonella* (20.5%) demonstrated resistance to more than one antimicrobial agent, whereas *Shigella* isolates did not exhibit resistance to more than one drug. All *Shigella* isolates were sensitive to all discs except for Ampicillin. The isolates of *Salmonella* were 100% (39/39) resistant to ampicillin, while the isolates of *Shigella* were completely (39/39) sensitive to ceftriaxone (Table 4).

## Factors associated with enteric bacteria and intestinal parasites among food handlers

Following multivariable analysis to control for potential confounding variables, several factors were found to be significantly associated with the infection of enteric bacteria and intestinal parasites. Individuals who were unable to read and write demonstrated significantly higher odds of infection (AOR = 1.96, 95% CI: 1.18–3.26). Furthermore, not trimming fingernails was associated with substantially elevated odds of infection (AOR = 2.52; 95% CI: 1.20–9.55). Similarly, not washing hands with soap after using the toilet significantly increased the odds of infection (AOR = 2.27, 95% CI: 1.17–4.42). Lastly, the consumption of raw food was also identified as a significant risk factor, with an AOR of 1.63 (95% CI: 1.39–5.00)*) for infection (Table 5).

## Discussion

Food handlers play a significant role in the spread of infections at restaurants and bars where food is served to large numbers people because they come into close contact with food products. As a result, this study aimed to assess the

**Table 2. Shows the hygiene practices of food handlers among food and drinking establishments in Bule Hora town, Ethiopia, 2023.**

| S.no | Variable | Response | Frequency | Percentage |
|------|----------|----------|-----------|------------|
| 1 | How often do you wash your hands before food handling | Always | 107 | 28.5% |
| | | Sometimes | 268 | 71.5% |
| 2 | How do often you wash your hands after toilet | Always | 235 | 62.6% |
| | | Sometimes | 140 | 37.4% |
| 3 | Availability of detergents at hand washing facility | Yes | 78 | 20.7% |
| | | No | 297 | 79.3% |
| 4 | Did they cover hair during the inspection | Yes | 194 | 51.6% |
| | | No | 181 | 48.4% |
| 5 | Did they wear a gown during the inspection | Yes | 201 | 53.7% |
| | | No | 174 | 46.3% |
| 6 | Gown clean | Yes | 81 | 40.4% |
| | | No | 120 | 59.5% |
| 7 | Did they wear jewelry during the inspection | Yes | 193 | 51.6% |
| | | No | 182 | 48.4% |
| 8 | Trimming fingernail | Yes | 192 | 51.2% |
| | | No | 183 | 48.8% |
| 9 | Have sick leave while feeling discomfort | Yes | 293 | 78.0% |
| | | No | 82 | 22.0% |
| 10 | Money handled by a food handler | Yes | 247 | 65.9% |
| | | No | 128 | 34.1% |
| 11 | Eating raw foods | Yes | 156 | 41.9% |
| | | No | 219 | 58% |
| 12 | Overall practice | Good | 72 | 19.1% |
| | | Poor | 303 | 81.9 |

**Table 3. Prevalence of intestinal parasites in food handlers among food and drinking establishments in Bule Hora Town, Ethiopia 2023 (n = 147).**

| S/N | Parasite | Frequency | Percentage |
|-----|----------|-----------|------------|
| 1 | *Ascaris lumbricoides* | 71 | 51.1% |
| 2 | *Trichuris trichiura* | 29 | 20.9% |
| 3 | *Hook worm* | 13 | 9.4% |
| 4 | *Teania spp* | 10 | 7.2% |
| 5 | *E. histolytica/dispar complex* | 11 | 7.9% |
| 6 | *Giardia lamblia* | 5 | 3.6% |

prevalence of enteric bacteria, intestinal parasite, their antimicrobial resistance profile, and their determinant factors among food handlers. According to this study, 39.2% of food workers have intestinal parasites. This finding aligned with studies in Yabelo town (36.4%) [10] and Addis Ababa (32.4%) [18]. In contrast to the current study, a lower prevalence of intestinal parasites was documented in Aksum town (14.5%) [19], Chagni town (14.8%) [20], and Ghana (21.5%) [21]. In addition, Ibrid, Jordan 48% [22] and Sanliurfa, South eastern Anatolia 52.2% [23] had reported high intestinal parasite prevalence compared to the current findings. In this case, the disparity may result from differences in food handlers' sanitary habits, which are influenced by socio-demographic traits.

**Table 4. Antimicrobial susceptibility pattern of *Salmonella* and *Shigella* isolates among food handlers in Bule Hora town, Ethiopia.**

| Antibiotics | Susceptibility for *Salmonella* | | | Susceptibility for *Shigella* | | |
|---|---|---|---|---|---|---|
| | S (%) | I (%) | R (%) | S (%) | I (%) | R (%) |
| Ampicillin | 0 (0) | 0(0) | 39(100) | 0(0) | 0(0) | 11(100) |
| Ciprofloxacin | 36 (92.3) | 0(0) | 3(7.7) | 11(100) | 0(0) | 0(0) |
| trimethoprim-sulfamethoxazole | 33(84.6) | 0(0) | 6(15.3) | 11(100) | 0(0) | (0) |
| Ceftriaxone | 39(100) | 0(0) | 0(0) | 11(100) | 0(0) | 0(0) |
| Gentamicin | 35(89.7) | 0(0) | 4(10.1) | 11(100) | 0(0) | 0(0) |
| Doxycycline | 39(100) | 0(0) | 0(0) | 11(100) | 0(0) | 0(0) |
| Nalidixic acid | 27(69.2) | 0(0) | 12(30.7) | 11(100) | 0(0) | 0(0) |
| Tetracycline | 12(30.7) | 0(0) | 27(69.2) | 11(100) | 0(0) | 0(0) |

S = Susceptible; I = Intermediate resistance; R = Resistance

**Table 5. Multivariate analysis for factors associated with the prevalence of intestinal parasites or enteric bacterial infections among food handlers in Bule Hora town, Ethiopia, 2023 (n = 375).**

| Variables | | Intestinal parasite or bacterial | | COR (95%CI) | AOR (95%CI) |
|---|---|---|---|---|---|
| | | Positive | Negative | | |
| Sex | Male | 48 | 42 | 1.07(0.48-3.304) | 0.709 (.433 −1.162) |
| | Female | 148 | 136 | 1 | 1 |
| Educational status | Unable to read and write | 102 | 80 | 1.31 (1.15- 6.11) | 1.963(1.182-3.26)* |
| | able to read and write | 95 | 98 | 1 | 1 |
| Training on food handling | No | 113 | 75 | 1.84(1.104-3.14) | 1.045(0.658-1.659) |
| | Yes | 84 | 103 | 1 | 1 |
| Hand washing with soap after using toilet | No | 92 | 48 | 2.37(1.46-5.56) | 2.27 (1.17 - 4.42)* |
| | Yes | 105 | 130 | 1 | 1 |
| Finger nail trimming | No | 113 | 70 | 2.07(1.46-10.61) | 2.52 (1.205 −9.55)* |
| | Yes | 84 | 108 | 1 | 1 |
| License status | Not Licensed | 60 | 52 | 1.06(1.87-5.98) | 0.929(.571 −1.512) |
| | licensed | 137 | 126 | 1 | 1 |
| Eating raw foods | Yes | 100 | 56 | 2.24(1.85-5.99) | 1.632(1.39-5.001)* |
| | No | 97 | 122 | 1 | 1 |
| Wearing jewelry | Yes | 107 | 86 | 1.25(0.25-8.91) | 1.1(0.67-5.78) |
| | No | 90 | 92 | 1 | 1 |

*-variable with significant association

From intestinal parasites the most prevalent was *Ascaris lumbricoides* (51.1%) in a recent study. This finding was consistent with studies carried out in Yabelo [24]. However, the most common intestinal parasite is *Entamoeba histolytica,* according to studies done in Chagni Town [20], Addis Ababa [25], Nekemte Town [26], and Nigeria [27]. The disparity in intestinal parasite incidence among regions could be caused by variations in personal cleanliness habits, routine health examinations, socio-demographic traits, identification laboratory reagents, and geographic dispersion.

In a recent study, the prevalence of enteric bacteria (*Salmonella* and *Shigella)* was found 13.3%. This finding is in line with research conducted in Yabelo town [24] and Arbaminch (9.5%) [28] and higher than that of studies conducted at Debre Markos (5.9%) [29] and Haramaya University (5.04%) [30]. But it's not as high as Research in Abuja, Nigeria,

showed that *Shigella* and *Salmonella* were more common(57.8%) [27]. This might be resulted from variation in WASH facilities, variation in food safety practices and socioeconomic status and population density.

In the current study, the *Shigella* isolate was 100% sensitive to doxycycline, ciprofloxacin, ceftriaxone, gentamicin, and trimethoprim-sulfamethoxazole. This finding agrees with findings from Gondar town [30], Motta town [29], Arbaminch University [31], and Addis Ababa [32]. In addition, the isolates' antimicrobial resistance profiles showed that Salmonella was 100% resistant to ampicillin, 69.2% resistant to tetracycline, 30.7% resistant to nalidixic acid, 16.3% resistant to trimethoprim-sulfamethoxazole, 10.1% resistant to gentamicin, and 7.7% resistant to ciprofloxacin. Antimicrobial tests revealed that the remaining isolates were susceptible. Studies conducted in Yabelo [24], Addis Ababa [25], and Dire Dawa City, Ethiopia [29], were in agreement with this result. Differences in sensible drug use may be the cause of the variances in sensitivity and resistance.

In a recent study, only 19.1% of food handlers had good hygienic practices. This study is lower than the ones conducted in Addis Ababa (27.4%) [25], and Gonder University cafeteria (46.7%) [30]. This variation will be due to variations in socio-demographic characteristics and study periods. The hygienic practices of food handlers might have a direct relationship with their health status. This highlights an urgent need for improved training and adherence to safety protocols

In the present study, the multivariable analysis reveals that food handlers who eat raw vegetables, fruits, and meats are 1.62 times more likely to be positive for intestinal parasites and bacteria. This study is similar to a study conducted in Yabello, Ethiopia [24]. Additionally, the food handler who was unable to read and write was 1.96 times more likely to be positive for intestinal parasites and bacteria. This study's findings are also similar to those of a study conducted in Aksum town [31]. Those unable to read and write will be challenged to understand the food handler's safety practices.

Furthermore, in the current study, regular hand washing with soap after the toilet has a significant association with the health status of food handlers. The odds of being positive for intestinal parasites and bacteria were 2.27 times higher among those who didn't regularly wash their hands after the using toilet compared to others. This study's findings are similar to those of the study conducted in a different part of the country, in Ethiopia, and Kano Metropolis, Kano State, Nigeria [27]. Regular hand washing with soap can reduce intestinal parasites and bacteria by removing or killing them since the soap has an antimicrobial impact. Thus, regular hand washing after the toilet is recommended to reduce the prevalence of parasites and bacteria, especially in food and drinking establishments where food is served to a large number of people.

Besides, regular finger trimming has a significant association with the health status of food handlers. The odds of being positive are 2.52 more likely among those who do not trim their fingernails regularly. This finding is similar to the study conducted in Nekemte town, Ethiopia [26], and Changi town, Ethiopia [20]. Regularly untrimmed fingernails can harbour different intestinal parasites and bacteria and contaminate food and drink when food handlers prepare and serve food for consumption.

One of the weaknesses of this study was being cross-sectional, in which the temporal relationship between cause and exposure cannot be established and lack of molecular studies; however, a Formal ether concentration procedure was followed in laboratory analysis, and an observational checklist was used in addition to self-report by food handlers. This study finds out the antimicrobial resistance profile of enteric bacteria and parasites, which is the current issue.

## Conclusion

In the study are Intestinal parasites, *Salmonella* and *shigella* remains public health problem. The majority of the *Salmonella* and *Shigella* isolates were sensitive to the tested antimicrobials. Educational status, eating raw vegetables or fruit, hand washing after the toilet, and fingernail trimming have associations with the health status of food handlers. It recommended strengthening hygiene education, and handing washing practice.

## Acknowledgments

First, we would like to glorify almighty God for his endless support. Next, our deepest gratitude goes to Bule Hora University for its facilitation in this study. Lastly but not least, we would like to thank food and drinking establishment managers and food handlers for their willingness to participate in this study.

## Author contributions

**Conceptualization:** Lechisa Asefa.

**Data curation:** Lechisa Asefa.

**Methodology:** Lechisa Asefa, Anteneh Fikirie.

**Resources:** Lechisa Asefa.

**Validation:** Anteneh Fikirie.

**Writing – original draft:** Lechisa Asefa, Gudeta Kumela, Habtamu Roba, Angefa Ayele, Gedeno Karbana, Degefa Dhengesu, Hailu Lemma.

**Writing – review & editing:** Lechisa Asefa, Gudeta Kumela, Habtamu Roba, Angefa Ayele, Gedeno Karbana, Degefa Dhengesu, Hailu Lemma, Anteneh Fikirie.

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
