## [Decision Letter · Decision Letter 0]

2 Dec 2025

Dear Dr. Asefa,

Thank you for submitting your manuscript to PLOS ONE. After careful consideration, we feel that it has merit but does not fully meet PLOS ONE’s publication criteria as it currently stands. Therefore, we invite you to submit a revised version of the manuscript that addresses the points raised during the review process.

Please especially address concerns raised by Reviewers 2 and 3 regarding lack of clarity and providing additional details regarding laboratory methodologies.

We look forward to receiving your revised manuscript.

Kind regards,

Timothy J Wade, Ph.D

Academic Editor

PLOS ONE

Journal Requirements:

2. In the online submission form, you indicated that [data will be avaialble based on request].

Reviewers' comments:

Reviewer's Responses to Questions

**Comments to the Author**

1. Is the manuscript technically sound, and do the data support the conclusions?

Reviewer #1: Yes

Reviewer #2: Partly

Reviewer #3: Yes

2. Has the statistical analysis been performed appropriately and rigorously?

Reviewer #1: Yes

Reviewer #2: Yes

Reviewer #3: Yes

3. Have the authors made all data underlying the findings in their manuscript fully available?

Reviewer #1: Yes

Reviewer #2: Yes

Reviewer #3: Yes

4. Is the manuscript presented in an intelligible fashion and written in standard English?

Reviewer #1: Yes

Reviewer #2: No

Reviewer #3: Yes

Reviewer #1: In this manuscript, the authors have presented a well written and detailed study that concluded that there is a high prevalence of parasites and bacteria among food handlers in the given study area. This cross-sectional study conducted in Bule Hora town investigated the prevalence of intestinal parasites and enteric bacteria, as well as their antimicrobial resistance profiles, among 375 food handlers. The study aimed to address a notable knowledge gap in effective public health interventions in the region. Increased educational status, improved hygienic practices (fingernail trimming, handwashing), and better dietary habits can significantly influence their overall health status.

Reviewer #2: The study by Asefa et al, entitled "Microbial threats: Magnitude of Intestinal Parasites, Enteric Bacteria, and Antimicrobial Resistance Profile and Associated Factors among Food Handlers, West Guji Zone, Ethiopia" is relevant to public health and microbiology, especially in regions with a high burden of foodborne diseases.

The focus on intestinal parasitosis, Salmonella, Shigella, and antimicrobial resistance is appropriate for PLOS ONE.

However I have some important comments:

1) The title suggests a broad focus (“Microbial threats”), but the study only analyzes two bacteria (Salmonella and Shigella) and intestinal parasites.It is recommended to adjust the title or justify why only these agents were considered.

2) The introduction needs more microbiological argumentation on: Why Salmonella and Shigella are priorities in the area; The global and local importance of antimicrobial resistance; Previous models or evidence of risk in food handlers in Ethiopia.

3) There are onconsistencies detected:

-Inconsistency of percentages and counts in several parts (e.g., the abstract mentions 39.2% and then 52.5% as the total prevalence).

-Variables are sometimes mixed without clarifying whether they are independent or confounding.

- The wording in Results sometimes mixes descriptive observations with interpretive analysis (which should go in Discussion).

- In some paragraphs, the term “health status” is used interchangeably to refer to Parasitic infection, Bacterial infection and Hygiene practices; This is conceptually incorrect and confusing.

4) Key references on microbiology and antimicrobial resistance of Salmonella/Shigella are missing; please revise WHO Global Antimicrobial Resistance Surveillance System (GLASS), Current CLSI M100 (2023–2025), Regional epidemiological data from FAO/WHO, Some articles cited are duplicates or irrelevant to microbiology, Several references are incorrectly cited (format, incomplete information).

5) There are also some methodological issues:

- Lack of clarity regarding what method was used to identify E. histolytica/dispar? (E. histolytica cannot be identified by microscopy alone); "Bacteria” seems to refer exclusively to Salmonella and Shigella, but the text uses “bacteria” as if it were general, this should be explicitly clarified.

- Samples for bacteria, transport in Cary Blair at 4°C is correct, but the maximum time allowed according to CLSI should be indicated. It is not specified whether late samples were discarded.

- Antimicrobials tested, the panel is limited and does not include antibiotics recommended by CLSI for Salmonella/Shigella (Azithromycin, Cefotaxime/cefixime, Carbapenems (in surveillance studies), Doxycycline is not a drug recommended for Shigella by CLSI.

- Lack of molecular methodology, it is desirable molecular confirmation (PCR) of serotypes in Salmonella and testing for common resistance genes (such as blaTEM, gyrA, etc.).

6) The results are useful, but not sufficient for broad conclusions such as “high prevalence” or “public policy recommendations.” The combined prevalence (52.5%) mixes parasites + bacteria, which is not a valid epidemiological metric. Also, there ar not presented distribution by age/sex for each agent. Could be useful bar charts or trend graphs. You can also include a heatmap for Parasite load (intensity), only presence/absence. Regarding the antimicrobials there is a lack of presentation of mean halos or standard deviation and no confirmation of whether current CLSI cut-off points were applied.

7) The discussion section shows important issues:

- Repeats many results instead of interpreting them.

-Does not discuss the actual microbiological impact of Salmonella/Shigella on food handlers; the implications for global antimicrobial resistance; nor the limitations of parasite diagnosis by microscopy and study biases. Does not explain why Shigella is 100% sensitive (unusual data requiring serious microbiological interpretation) and does not analyze whether the associated variables have a plausible causal model.

8) The conclusions are partialy coherent, but stating that prevalence is “high” is subjective without a point of comparison or epidemiological criteria. The recommendations are not directly derived from data (e.g., knowledge of hygiene and the impact of training were not measured). Key limitations are not mentioned (only one in Discussion).

Reviewer #3: I have already included my comments in track change to the main text. It would have been also nice to explain the laboratory procedures performed in detail especially for the parasite ID. The issue of participants eligibility to the study was not addressed.

**Do you want your identity to be public for this peer review?** For information about this choice, including consent withdrawal, please see our Privacy Policy

Reviewer #1: No

Reviewer #2: **Yes:** Edwin Barrios Villa

Reviewer #3: No

---

## [Author Response · Author response to Decision Letter 1]

22 Dec 2025

Response letter

We are pleased to submit the revised version of our manuscript, together with the responses to the comments. We would like to thanks for the constructive comments, which truly improved the level of the manuscript. We believe that we addressed all the comments/suggestion. After careful consideration of the comments, we have outlined point by point response together with the comments (copy and pasted), which is also uploaded as separated file. The specific changes in the main text are highlighted in yellow color. We hope the revised manuscript will received your favorable consideration for publication

Response for Reviewer 2

1) The title suggests a broad focus (“Microbial threats”), but the study only analyzes two bacteria (Salmonella and Shigella) and intestinal parasites.It is recommended to adjust the title or justify why only these agents were considered.

Response: - Than you for your comment we have adjust title to “Magnitude, Resistance Profiles, and Risk Factors of Intestinal Parasites and Enteric Bacteria among Food Handlers in West Guji Zone, Ethiopia”

2) The introduction needs more microbiological argumentation on: Why Salmonella and Shigella are priorities in the area; The global and local importance of antimicrobial resistance; Previous models or evidence of risk in food handlers in Ethiopia.

Response: - Thank you for your valuable comment. I have incorporated the suggested information into the main manuscript.

3) There are onconsistencies detected:

-Inconsistency of percentages and counts in several parts (e.g., the abstract mentions 39.2% and then 52.5% as the total prevalence).

Response: - Thank you for your comment what we wanted to say was the prevalence of both bacterial and parasite (52.5%). And 39.2% is the prevalence of parasite alone. However based on your comment as well as other reviewer comment we have removed sentence which is prevalence of both bacterial and parasite (52.5%).

- The wording in Results sometimes mixes descriptive observations with interpretive analysis (which should go in Discussion).

Response: - Thank you for your best comment I have removed them from result section

- In some paragraphs, the term “health status” is used interchangeably to refer to Parasitic infection, Bacterial infection and Hygiene practices; This is conceptually incorrect and confusing.

Response:-Thank you for your excellent insight. In this study, we assessed the health status of food handlers by examining both bacterial and parasitic infections, as well as hygienic practices to identify potential risk factors. I have reviewed my document and could not find any instance where these variables were used interchangeably. They were analyzed independently, and they are independent and we can’t use interchangeably as you commented. Please can you located where it was happened?

4) Key references on microbiology and antimicrobial resistance of Salmonella/Shigella are missing; please revise WHO Global Antimicrobial Resistance Surveillance System (GLASS), Current CLSI M100 (2023–2025), Regional epidemiological data from FAO/WHO, Some articles cited are duplicates or irrelevant to microbiology, Several references are incorrectly cited (format, incomplete information).

Response:-Thank you I have suggestion I have incorporate it

5) There are also some methodological issues:

- Lack of clarity regarding what method was used to identify E. histolytica/dispar? (E. histolytica cannot be identified by microscopy alone); "Bacteria” seems to refer exclusively to Salmonella and Shigella, but the text uses “bacteria” as if it were general, this should be explicitly clarified.

Response: - Thank you for your critical comment for parasite identification. Thus in our case it should be E. histolytica/dispar complex. We accepted and updated through the document. Besides, we have edited issue of using general term “bacteria” with Prevalence of salmonellosis and shigellosis

- Samples for bacteria, transport in Cary Blair at 4°C is correct, but the maximum time allowed according to CLSI should be indicated. It is not specified whether late samples were discarded.

Response: - This is very great concern. The sample was transported within one hour of collection and analyzed within 48 hours.

- Antimicrobials tested, the panel is limited and does not include antibiotics recommended by CLSI for Salmonella/Shigella (Azithromycin, Cefotaxime/cefixime, Carbapenems (in surveillance studies), Doxycycline is not a drug recommended for Shigella by CLSI.

Response: - This is very great concern. The comments is accepted

- Lack of molecular methodology, it is desirable molecular confirmation (PCR) of serotypes in Salmonella and testing for common resistance genes (such as blaTEM, gyrA, etc.).

Response: - This very great suggestion. We couldn’t afford and get access to PCR for our study and it may be included under limitation of the study.

6) The results are useful, but not sufficient for broad conclusions such as “high prevalence” or “public policy recommendations.” The combined prevalence (52.5%) mixes parasites + bacteria, which is not a valid epidemiological metric. Also, there ar not presented distribution by age/sex for each agent. Could be useful bar charts or trend graphs. You can also include a heatmap for Parasite load (intensity), only presence/absence. Regarding the antimicrobials there is a lack of presentation of mean halos or standard deviation and no confirmation of whether current CLSI cut-off points were applied.

Response: - Thank you for your comment. Conclusion was edited. I have removed combining bacteria and parasite prevalence. However, we haven’t presented distribution by age/sex for each agent group but we have make sex/age independent variable which may affect the health outcome of food handles. For AMS test the CLSI cut-off points were applied.

7) The discussion section shows important issues:

- Repeats many results instead of interpreting them.

Response: - some key finding was written on discussion for reader to easily understand it. We accept comment on interpretation of result and tried to interpret the results in main documents.

-Does not discuss the actual microbiological impact of Salmonella/Shigella on food handlers; the implications for global antimicrobial resistance; nor the limitations of parasite diagnosis by microscopy and study biases. Does not explain why Shigella is 100% sensitive (unusual data requiring serious microbiological interpretation) and does not analyze whether the associated variables have a plausible causal model.

Response: This is very interesting comment we tried to improve in main document

8) The conclusions are partialy coherent, but stating that prevalence is “high” is subjective without a point of comparison or epidemiological criteria. The recommendations are not directly derived from data (e.g., knowledge of hygiene and the impact of training were not measured). Key limitations are not mentioned (only one in Discussion).

Response: - Thank you for your comment. The conclusion and recommendation was updated in main document. To my best knowledge limitation is written in discussion section this why we made it in discussion. However in updated main document we add one limitation not discussed before.

Response for Reviewer 3

1. Introduction section, Re-write it….. use epidemiological term rather than using word dangerous..

Response: - This very interesting suggestion I have changed term to potential source of infection.

2. What did you do for the positive cases? Intervention taken has to be explained here.

Response: - This very interesting question. Food handlers with positive result were linked to Bule hora University teaching hospital for medication and health education was given on personal hygiene. This was done by maintain the privacy of the participants.

3. Would have better include the parasites, bacterias and co-infection with different parasites or with the bacteria including the table. How about the co-infection status?

Response: - Thank you for your suggestion. However, during laboratory analysis we haven’t considered it. So, this time we have no data on hand.

4. Italicize all parasites and bacterial names across the document

Response: - Thank you for your comment. We tried to make italic

5. How do the laboratory reagents be a factor for the differences? Did you see the diagnostic techniques employed?

Response: Oh! Thank you for good view. We have removed this justification.

---

## [Editor Report · Decision Letter 1]

26 Jan 2026

Magnitude, Resistance Profiles, and Risk Factors of Intestinal Parasites and Enteric Bacteria among Food Handlers in West Guji Zone, Ethiopia

PONE-D-25-34546R1

Dear Dr. Asefa,

We’re pleased to inform you that your manuscript has been judged scientifically suitable for publication and will be formally accepted for publication once it meets all outstanding technical requirements.

Kind regards,

Timothy J Wade, Ph.D

Academic Editor

PLOS One
---

## [Editor Report · Acceptance letter]

PONE-D-25-34546R1

PLOS One

Dear Dr. Asefa,

I'm pleased to inform you that your manuscript has been deemed suitable for publication in PLOS One. Congratulations! Your manuscript is now being handed over to our production team.

Kind regards,

on behalf of

Dr. Timothy J Wade

Academic Editor

PLOS One